# A2 Milk: New Perspectives for Food Technology and Human Health

**DOI:** 10.3390/foods11162387

**Published:** 2022-08-09

**Authors:** Salvador Fernández-Rico, Alicia del Carmen Mondragón, Aroa López-Santamarina, Alejandra Cardelle-Cobas, Patricia Regal, Alexandre Lamas, Israel Samuel Ibarra, Alberto Cepeda, José Manuel Miranda

**Affiliations:** 1Laboratorio de Higiene Inspección y Control de Alimentos, Departamento de Química Analítica, Nutrición y Bromatología, Universidad de Santiago de Compostela, 27002 Lugo, Spain; 2Área Académica de Química, Universidad Autónoma del Estado de Hidalgo, Carretera Pachuca-Tulancingo Km. 4.5, Pachuca 42076, Hidalgo, Mexico

**Keywords:** A2 milk, β-casein, β-casomorphin 7, lactose intolerance, gastrointestinal discomfort

## Abstract

Although milk consumption is increasing worldwide, in some geographical regions, its consumption has persistently declined in recent decades. This fact, together with the increase in milk production prices, has caused both milk producers and the dairy industry to be immersed in a major crisis. Some possible solutions to this problem are to get people who do not currently consume milk to start drinking it again, or to market milk and dairy products with a higher added value. In this context, a type of milk called A2 has recently received attention from the industry. This type of milk, characterized by a difference in an amino acid at position 67 of the β-casein polypeptide chain, releases much smaller amounts of bioactive opioid peptide β-casomorphin 7 upon digestion, which has been linked to harmful effects on human health. Additionally, A2 milk has been attributed worse technological properties in the production of some dairy products. Thus, doubts exist about the convenience for the dairy industry to bet on this product. The aim of this review is to provide an update on the effects on human health of A2 milk, as well as its different technological properties to produce dairy products.

## 1. Introduction

Milk is regarded as one of the staples of Western diets because of its high nutritional value. It is the first food in the diet of mammals, providing all the energy and nutrients necessary for growth and development in their first periods of life [1]. Milk intake stops after weaning in all mammals except in humans, who continue their consumption in adulthood, not only as milk but also as dairy products. In recent years, the dairy sector has been immersed in a major crisis in Europe due to high price volatility, increments of production costs, and a recent deregulation process, due to the abolition of milk quotas [2,3]. Additionally, their consumption is being replaced in some consumer groups by plant-based milk substitutes, which are often presented as a healthier, more sustainable, and animal-friendly alternative to bovine milk [1].

In this context, the dairy sector needs to find ways to increase profitability [2]. One possible alternative is to incorporate milk into the diet of consumers who do not currently consume it. In this case, one of the major segments of the population that could increase the consumption of milk and dairy products are those who suffer adverse reactions after ingestion [4]. Digestive discomfort is mainly associated with lactose malabsorption, which affects approximately 65% of the adult population worldwide [5]. Lactose-intolerant individuals suffer diverse digestive symptoms after milk ingestion, such as abdominal pain, bloating sensation, stool frequency changes, and stool consistency changes [6]. Another option to increase dairy profitability is the search for differentiated products with a higher added value [7], such as milk or dairy products with associated benefits to human health. Food with health claims shows higher prices than those observed in high-volume food segments [8].

More than 95% of the proteins contained in cow’s milk consist of caseins and whey proteins. Caseins represent approximately 80% of total milk protein, and whey proteins represent about 20%. Among bovine skim milk caseins, four different types have been described: α-casein S1 (ranging 12–15 g/L), β-casein (9–11 g/L), α-casein S2 (3–4 g/L), and κ-casein (2–3 g/L). Among other protein fractions in bovine skim milk, the most relevant content is for α-lactalbumin (0.6–1.7 g/L), serum albumin (0.4 m/L), immunoglobulins (0.5–0.8 g/L), lactoferrin (0.02–0.1 g/L), and the secretory component (0.02–0.1 g/L) [9].

These proteins have a major implication in milk production, the industry, and consumer health. β-casein occupies the second position in bovine milk caseins in terms of abundance, in addition to presenting many amino acids [10,11]. All types of caseins undergo modifications in their structures due to the substitution or exclusion of some amino acids of the peptide chain, thus generating genetic variants. When there is more than one structural variant encoded by a gene, it can be referred to as a polymorphism [12]. These genetic variants that affect protein structures cause changes in their characteristics, in addition to influencing milk production and technological applications for industrial use [13].

In recent years, a new type of cow’s milk, named “A2 milk”, has been introduced in the market. This type of milk was first commercialized in New Zealand and has since been gaining a presence in the markets of several countries [7]. A2 milk is characterized by being free of the A1 variant of β-casein: a protein that represents approximately 30% of the caseins in cow’s milk [8]. By genotyping dairy cows and using semen from selected bulls, or genotyping and selecting only A2 calves for replacement, it is possible to obtain a dairy farm producing only A2-type milk [7]. The coding gene in β-casein synthesis is the CNS2 gene, for which 13 different allelic variants have been described (A1, A2, A3, B, C, D, E, F, G, H1, H2, and J) [14]. The A2 variant is present in the milk of many mammals, both in humans and in goats, sheep, and cows, while the A1 variant is only present in cattle [12].

The two most common variants of this protein are types A1 and A2, which differ by only one amino acid at position 67. In type A1 β-casein, there is an amino acid histidine at this position, whereas in type A2 protein, this histidine is replaced by a proline [15]. Thus, the original codon cytosine-cytosine-thymine (CCT), which forms the amino acid proline in the A2 variant, is modified to cytosine-adenine-thymine (CAT), which encodes the formation of histidine at position 67 of the β-casein polypeptide chain in A1 variants [16]. Dairy cow breeds show different β-casein patterns in their milk. Thus, the breed with the highest percentage of A2 alleles in Europe is Guernsey, with 92%, whereas A1 is the most common allele in other dairy breeds, such as Holstein (60%) or Ayrshire (60%) [17]. Currently, most milk marketed contains a mixture of A1 and A2 β-casein, which may be from A1/A2 heterozygous cows or from the milk mixture of A1/A1 and A2/A2 homozygous animals [18].

## 2. The A2 Milk Market

Although the worldwide consumption of milk and dairy products is constantly increasing and is expected to continue increasing over the next decade [19], milk consumption has decreased significantly since the 1970s in specific geographical areas, such the United States and the European Union [20,21]. Thus, the dairy industry has tried to be creative and develop new products to increase consumption. In 2003, the A2 Milk Company Limited emerged in New Zealand, commercializing both milk and dairy products (cheeses, yogurts, or creams) free of the A1 variant of β-casein. A2 milk strongly entered the market in this country and covered almost 10% of the milk market in Australia. In view of the possible benefits of A2 milk for human health, in addition to avoiding the negative effects of β-casein A1, many farmers around the world have switched to A2 milk production [22]. This successful market trend has spread to other geographic areas, such as North America, Europe, and China [20]. Consequently, other companies dedicated to the commercialization of semen for dairy farms have introduced the A2/A2 genotype in their sire directories as a characteristic of interest and added value for their animals [23].

The adoption of the A2 milk production system is complicated because not all farmers are willing to genotype their cattle to obtain 100% A2 milk. Generally, it is assumed that the Al allele brings better production performance characteristics in dairy cows than the A2 genotype [24], although some works did not find significant differences in milk production [25], and some studies even found that higher milk production rates are obtained with the A2 genotype [11,26]. Economically, genetic selection is not an excessive expense because genotyping tests are becoming cheaper [23]. However, the milk production process must be separated from that of milk coming from A1 cows. The cows must be milked separately, in separate tanks, with milk independently transported, brought to the factory separately, and manufactured without mixing or contaminating with A1 protein, as the risk of this happening is very high. Thus, it is necessary to have a large potential market and a milk price premium to make the whole process profitable. Many people could be interested in consuming only A2 milk, but only if its beneficial effects for human heath have been demonstrated by scientific evidence [7]. A recent work investigating consumers’ awareness of A2 milk [21] revealed that the factors that condition consumers to buy these dairy products are price; origin; and quality certification, such as “organic” or “traditional”. This same work revealed that the premium price that Italian consumers are willing to pay for A2 milk with respect to fresh lactose-free milk is approximately 20-euro cents/liter [21]. Other work [27] revealed that only 38% of a Brazilian group of consumers would pay an extra price for A2 milk with respect to conventional milk. The first dairy company commercializing A2 milk (The A2 milk company) claims that they pay a premium of around 5–7% to its farmer suppliers in New Zealand, Australia, and the United Kingdom [28].

A recent work [29] revealed that a high proportion of consumers did not know about A2 milk, and thus had never considered purchasing this type of milk. Moreover, a positive relationship was found between socioeconomic level and A2 milk awareness [30]. Therefore, it seems important to promote marketing strategies to improve awareness of A2 milk among less informed consumers and modify their buying behavior.

Additionally, in recent years, some legal problems have arisen from the adulteration of some milk marketed as A2 with other non-A2 milks. This addition of non-A2 milk to A2 milk represents fraud for consumers because they pay an extra price for this adulterated A2 milk, and they may be concerned about the consumption of mixed β-casein variant A1 [22]. This fact makes it necessary for the marketing of A2 milk at a premium price to be viable and for consumer authorities to implement good systems to control this possible fraud.

## 3. Physical and Technological Properties of A2 Milk

A recent work [31] investigating the sensory quality, color, and composition of A2 milk compared with A1 milk found that different genotypes did not affect the smell, taste, or general acceptance of the milk. However, some differences were found in the color. A2 milk showed color parameters closer to the gold standard for color, making it more appealing to consumers without artificial food coloring [31].

Milk protein composition is an important factor in the nutritional and technological characteristics of milk. The quantities and proportions of milk caseins and whey proteins have been identified as playing a major role in milk coagulation and curd firming processes [32]. Several studies have extensively assessed the effects of milk protein genetic variants on the composition and protein profile of milk [31,32,33] and its coagulation properties [31,32,33,34,35]. Milk caseins play an important role in the production of dairy products, such as in cheese manufacturing [36,37]. However, there are still few reports about the influence of A2 milk on dairy product manufacturing and consumer perceptions.

Because of the increasing trend in A2 milk consumption in some countries [38,39,40], some farmers and breeding associations began to increase the frequency of the β-casein A2 allele in dairy cattle populations without paying attention to the potential effect of this change on milk technological characteristics [29]. Nevertheless, it was revealed that due to their composition, A2 milk has a higher percentage of total proteins and fat than A1 milk [12]. A recent work found slight differences in the amino acid composition of A2 and A1 milk, showing that A2 milk had a higher amount of leucine that A1 milk but lower overall amino acid content [31]. A2 milk could also have worse technological characteristics to produce some dairy products, such as cheese or yogurt [29]. Therefore, producers should keep in mind that if its production is not for sale as drinking milk but as an ingredient for the manufacturing of dairy products, the better price they expect to obtain for A2 milk could remain without effect or even be considered a negative factor by the industry.

β-casein influences the gelation and coagulation of milk because it plays a central role in casein micelle formation. According to Nguyen et al. [39], A2 β-casein is less hydrophobic, more soluble, and has a higher chaperone activity than A1 β-casein. According to this, several authors [31,32,33,34,35,36,37,38,39,40,41,42,43] associated the A2 milk variant with poor rennet coagulation properties. A further study [29] also found that the A1 milk genotype was associated with the best cheese-making abilities with respect to A2 milk due to a slight worsening of the coagulation properties for A2 milk.

Other works [38] suggested that different genetic variants of β-casein could influence the distribution and balance of calcium in milk. Thus, such genetic variants could influence the assembly and structure of casein micelles [43], thereby affecting curd formation during cheese making (Table 1). Several authors [38,39,40,41,44] reported that the gelling process of A2 milk is slower during fresh cheese making. In addition, a more porous microstructure and finer protein strands were observed in the gel made from milk with β-casein A2, which cause a lower gel strength compared to the gel from β-casein A1 milk [38]. However, a group of consumers detected no differences at the organoleptic level when asked about the taste of two types of Brazilian cheeses made from A1 or A2 milk, and only in one case (Minas Frescal cheese) did consumers find the cheese made from A2 milk softer and creamier and less consistent, rubbery, and drier than A1 cheese [27]. From these results, it can be concluded that A2 milk is associated with higher digestibility and better gut physiology, and the dairy industry should favor this genotype in milk destined for fresh consumption. However, it is inadvisable to use milk containing only the A2 variant of β-casein for cheese production, as this is associated with a worsening of milk technological characteristics and, consequently, a less efficient cheese-making process [29].

Additionally, it was reported that A2 milk had lower fat globule diameters [45] and higher polyunsaturated fatty acids content [31] than A1 milk [45]. Milk fatty acids and fat globule size influence the physicochemical, nutritional, and sensorial properties of milk and milk products [45]. The morphometric characteristics and fatty acid composition of milk are strongly influenced by casein polymorphism, as shown by the significant differences found among casein haplotypes of milk. These results are of interest because the degree of differentiation in globule size influences renneting, cheese texture, color, flavor, and butter texture. With respect to fat % in A2 milk, some authors found higher rates of fat in A2 animals than in A1 genotype cows when produced by the same dairy cow breed and in the same ambient [11,24].

Other important processes for the dairy industry for which a different activity was reported from A1 and A2 milk are emulsion and foaming capacities, although the results reported are not entirely consistent. The A2 β-casein variant was associated with poorer foaming capacity compared to A1, which was due to a more extensive spread of β-casein A1 at the interface, facilitating the more rapid formation of a coherent interfacial layer than in A2 [46]. In contrast, it was also reported that milk with the β-casein A2 variant had better foaming properties than A1 milk [37]. The emulsion properties of milk containing different β-casein variants were compared by Darewicz and Dziuba [47]. They concluded that A2 milk was more efficient than A1 milk in the emulsion formation, but its emulsions were less stable than those formed with the A1 and B variants. In addition, the A1 and B variants have better ordered structures in the absorbed state than A2, which also contributes to differences in their emulsifying ability [47] (Table 1).

In addition to cow’s milk, one work also investigated the allergenic and physicochemical properties of A2 goat’s milk [48], finding that the physicochemical properties of the A2 β-casein fraction are similar to those of bovine whole casein.

**Table 1 foods-11-02387-t001:** Differences in sensory characteristics and technological properties of A1 and A2 milk and dairy products.

Parameters Investigated	Samples	Main Findings	Reference
Chemical, protein profiles and rheological properties of milk and yogurt	Dried milk power obtained from raw A1A1 and A2 whole milk from Kiwi crossbred cows	- A2 milk had higher free calcium concentration and better foaming formation than A1 milk. - Yogurt prepared with A2 milk capacity had better porous microstructure and thinner protein strands than A1 milk-added yogurt.	[37]
Triangular test, focus group test, temporal dominance of sensations, overall sensory acceptance, online questionnaire with 17-multiple test questions	Petit Suisse and Minas Frescal cheese made with both A1 and A2 bovine milks	- Minas Frescal samples performed with A2 milk were softer and creamier than those made with A1 milk. - No difference in overall acceptability was found for Petit Suisse cheese. - Consumers reported that they did not read labels and the information about the type of milk frequently.	[27]
Dinamyc rheological analysis, rennet coagulation time, maximum coagulum strength, and curd firming rate	Morning milk obtained from 892 individual Danish cows (456 Holstein-Friesians and 436 Jerseys)	- A2 milk showed poorer coagulant properties than other tested milks.	[40]
Rheological analysis	Morning milk collected from 1299 Danish Holstein, Danish Jersey, and Swedish Red cows	- A2 milk showed poorer coagulant properties than other tested milks.	[42]
Emulsion properties, surface excess, conformational analysis, and MALDI-TOF spectra	Whole casein from milk obtained from individual Jersey, Friesian-Holstein, and German Black and White cow breeds	- β-casein B variant showed better emulsion-stabilizing properties than A2 and A1 variants.	[47]
Interfacial and foaming properties	Purified milk protein preparations	- A2 β-casein variant was associated with poorer foaming capacity compared to A1.	[46]
Composition, allergenic properties, and physicochemical properties	Goat milk	- The physicochemical properties of A2 β-casein fraction are similar to those of bovine whole casein.	[48]
Milk composition, cheese-making traits, and protein fraction identification	Individual milk samples collected from 1133 Holstein Friesian cows reared in 5 different herds	- A2 milk showed higher β-lactoglobulin and α-lactalbumin, as well as a lower production of β-casein with respect to the A1 milk. - Regarding milk cheese-making ability, the A2 genotype showed worst performance compared with the other genotypes, particularly with respect to the A1, with a higher rennet coagulation time.	[29]
Milk composition, and capillary electrophoresis	Morning milk samples and tissue/blood samples were collected from 415 dairy cows (20 Danish Holstein, 22 Danish Jersey, and 392 Swedish Red)	- Higher frequency of A1 milk, together with a decrease in A2 milk, could have positive effects on processing of cheese.	[32]
Composition analysis	23,970 milk samples from 2859 Holstein cows	- A2 milk showed worse milk coagulation time and curd firmness than B genotypes.	[35]
Fat globule size, fatty acids profile	250 Holstein were defined for their genotypes	- A2 milk showed most small fat globules and less big fat globules than other genotypes.	[45]
Protein characterization, rheological analysis	Individual milk samples from 121 cows in mid lactation of the Swedish Red and Swedish Holstein	- The β-casein A2 genotype was associated with inferior milk coagulation characteristics.	[44]
Quantification of milk proteins, casein micelle size, milk fat globule size, milk coagulation properties, salts distribution	Individual morning milk samples from 99 Norwegian Red cattle cows	- A1 milk showed better coagulation properties such as rennet coagulation properties and crud firmness than A2 milk.	[49]
Milk protein characterization	Morning milk samples and blood samples of 1912 firt-lactation Holstein-Friesian cows	- A2 genotype was associated with higher relative concentrations of β-casein, lower of αS2-casein, and with higher protein yield than A1 genotype.	[50]
Milk protein and fat characterization	Milk and blood samples of 20,928 Ayrshire cows	- Milk and protein production was highest for the β-casein A2 genotype, and fat percentage was highest for the A1 genotype.	[51]
Milk composition, physicochemical analysis, gelation properties	Milk from genotyped 114 cows	- A2 milk was associated with poor acid gelation properties.	[52]
Milk coagulation traits and protein composition	1042 multiparous Holstein cows	- A2 milk was instead associated with poorly coagulating milk, higher protein content, and made milk less suitable for cheese making than other genotypes.	[53]

## 4. Implications for Human Health of Dairy β-Caseins and β-Casomorphins

Protein composition explains many of the applications of milk in the industry, in addition to its nutritional value. Many milk protein characteristics are a consequence of genetic polymorphisms of dairy cows, which is why there are stakeholders to improve milk production in the dairy industry and to improve bovine herds [12]. When dairy products are consumed, digestive enzymes in the human intestine act on β-casein A1, thereby releasing the bioactive opioid peptide β-casomorphin 7 (BCM-7), which is primarily responsible for triggering the allergic process and is associated with a delayed milk gastrointestinal transit time [5,54]. The opposite is true for β-casein A2, which only under specific in vitro conditions with pH and enzymatic conditions not found in the human intestine can release small amounts of BCM-7 [54]. In fact, the BCM-7 concentration in the gut was found to be four times higher when produced from milk obtained from homozygous A1 cattle than from A2/A2 cattle [55]. Moreover, BCM-7 may be detected in the blood after the digestion of A1 milk but not after the consumption of A2 milk [13].

In dairy products, although BCM-7 has been detected in several types of cheeses, it is generally at lower concentrations than in milk [39]. Nguyen et al. [39] reported that BCM-7 is at a higher concentration in mold cheeses than in semihard cheeses because proteolytic enzymes from cheese starter cultures reduce its presence during ripening. The hydrolysis of BCM-7 by specific yogurt-forming bacteria reduced its presence to an unusual level [14].

Thus, after the intake of milk or dairy foods containing β-casein A1, the produced BCM-7 binds with µ-opioid receptors, which are responsible for pain or thirst, in the walls of the gastrointestinal tract and nervous system of consumers. In contrast, A2 milk consumption has been shown to increase the natural production of glutathione, which shows antioxidant activity and, consequently, health benefits. Due to its opioid-like activity, BCM-7 was proposed as a risk factor for adverse gastrointestinal symptoms perceived by consumers as milk intolerance, as well as for certain diseases [17], whereas A2 milk is considered safe.

Nevertheless, there is not convincing evidence for the relation of BCM-7 and consequently for β-casein A1 intake, and its adverse effects have been achieved by all authors [56,57]. Although related reviews have gathered a large amount of scientific evidence, the role of β-casomorphins (BCMs), such as BCM-7, and their physiological functions remain controversial, and more research with improved diagnostic techniques is needed to unequivocally determine their mechanism of action and study their possible health-related impacts [22]. Recently, other reviews [15,58,59,60] concluded that human-based published evidence provided moderate certainty for the adverse digestive health effects of A1 β-casein compared with A2 β-casein, but only indications and not conclusive evidence for the relation with other health effects. Globally, the adverse effects of BCM-7 on health have obtained positive results when experimental animal models were used for their study. Table 2 summarizes the results of the investigations about the effects of A1 or A2 β-casein in the development of gastrointestinal discomfort [61,62], gut microbiota [63], inflammatory response [61], type 1 diabetes [64,65,66], cardiovascular health [67], and pulmonary inflammation [68]. In all cases except one [64], no beneficial effect was found for A2 milk consumption with respect to A1 milk consumption.

Conversely, fewer positive effects of A2 milk in comparison to A1 milk consumption were found when trials were performed in humans (Table 3). Thus, only in the case of digestive intolerance did A2 consumption show beneficial effects when compared with A1 milk consumption [12,17,69,70], but no positive effects were found for cardiovascular markers [71,72], chronic functional constipation [73], muscle soreness [74], or type 1 diabetes [75]. With respect to neurological disorders, BCM-7 in the urine of autistic children who consumed A1 milk was found to be 10-fold higher than in children who consumed A2 milk, indicating a potential benefit of A2 milk consumption for this group [76].

As cited below, when A1 milk is consumed and reaches the human stomach, enzymatic digestion by gastric enzymes, such as pepsin, pancreatic elastase, and leucine aminopeptidase, would yield BCM-7 in an approximate proportion of 4 mg of BCM-7 released in the human jejunum from 30 g of A1 β-casein, which is related to increased inflammation and gastrointestinal discomfort [13,17]. This inflammation and discomfort derived from the digestion of A1 milk could explain the dairy intolerance that many individuals perceive as lactose or milk intolerance [13,16]. BCM-7 is also known to induce the production of mucins (the sticky proteins of mucus) [77], which provides a logical explanation as to why many people associate milk with mucus production.

This relationship was confirmed by a clinical trial comparing the effects of consuming milk with either β-casein A2 or a mixture of β-casein A1 and A2 for a 14-day period followed by an equivalent washout period in individuals who self-reported lactose intolerance. It was found that after consumption of the mixed A1/A2 milk, those individuals who self-perceived themselves as lactose intolerant experienced an exacerbation of lactose intolerance-related symptoms [13]. Even in consumers who are actually lactose intolerant, the substitution of A1 milk for A2 milk may ameliorate symptoms associated with intolerance. Clarke et al. [78] found in a rat trial that consumption of milk composed of 75% β-casein A2 elevated duodenal lactase activity compared to A1-fed rats. Subsequently, a randomized, double-blind clinical study found that A2 milk caused fewer symptoms of lactose intolerance in 25 subjects with maldigestion than mixed A1A2 milk [18]. Volunteers who consumed A1 milk reported abdominal pain, whereas those who consumed A2 milk showed no adverse effects [69].

BCM-7 may also influence lactose intolerance by affecting lactase production and activity, leading to hypolactasia with consequent symptoms of lactose malabsorption. On the other hand, BCM-7 was associated with gastrointestinal delay, which favors fermentation of lactose and many other oligosaccharides, causing many of the symptoms associated with lactose intolerance, such as colon irritation or inflammation, measured by the inflammatory marker myeloperoxidase [6].

Other negative effects attributed to A1 milk consumption have been widely discussed [16]. These potential beneficial effects of A2 milk consumption with respect to A1 milk are mainly related to allergic response, oxidative stress, diabetes, cardiovascular health, neurological disorders [4,16,79,80,81], sudden death infant syndrome [82], and the modulation of gut microbiota [63]. In addition to initiating gastrointestinal effects, BCM-7 released from A1 β-casein has been implicated in promoting oxidative stress by decreasing the uptake of the sulfur amino acid cysteine, which may be particularly important during the postnatal transition from placental to gastrointestinal nutrition in infants [78]. The adequate absorption of cysteine not only provides antioxidant resources for the gastrointestinal tract but also represents the portal to entry to support reduced glutathione (GSH) concentrations for the entire body. A recent clinical trial demonstrated that GSH concentrations in human subjects increased 2-fold after ingestion of A2 milk consumption compared with A1 milk consumption [83]. The health benefits of increased GSH levels are due to its high antioxidant capacity, which allows the aerobic metabolism to proceed without cellular damage caused by reactive oxygen species. The study revealed that supplementation with the A2 variant reduces the risk of diseases associated with oxidative stress and thus reduces the effects of aging, promotes the recovery of damaged tissues, and promotes fertility. Conversely, a decrease in GSH concentrations is associated with an increase in inflammation, including an increased release of proinflammatory cytokines such as tumor necrosis factor-alpha (TNF-α). BCM-7 can also exert a significant influence on methylation reactions, which is of particular importance for neurodevelopment [68,81].

### 4.1. Effects of A2 Milk on Allergies and Intolerance

The incidence of allergies through food consumption is low with respect to total allergies, not even representing 10% of the worldwide allergies that occur in the population under 18 years of age. The percentage of allergy occurrence has increased worldwide and is associated with numerous factors, such as race, age, and degree of urbanization [84]. Allergies associated with the consumption of BCM-7 have been gaining more importance in scientific studies. Different responses of infants to the consumption of breast milk and infant formulas have been observed; the latter causes alterations in bacterial adhesion and intestinal barrier status, as well as the dysfunction of tight junctions in the intestinal epithelium caused by calcium ions. On the other hand, BCM-7 elicits allergic responses not only in children but also in adults, since the presence of this peptide in the systemic circulation causes changes in the immune response [17]. The rationale for studying the difference between the A1 and A2 subvariants of the β-casein protein was based on the hypothesis that chronic functional constipation may in fact be an allergic immune response to the ingested BCM-7 protein [17].

BCM-7 can influence nervous, digestive, and immune functions through µ-opioid receptors located on the cell surfaces of these systems [85]. In this regard, Barnett et al. [61] compared the gastrointestinal effects of milk-based diets with β-casein type A1 or A2 in rats. They found a significantly longer gastrointestinal transit time in the A1 milk-fed group than for A2 milk, and that jejunal proline dipeptidyl peptidase IV enzyme activity was also higher in the A1 group than in the A2 group. Proline dipeptidyl peptidase IV is the main enzyme capable of hydrolyzing BCMs. In the same vein, in a study in mice, it was concluded that consumption of the β-casein A1 genetic variant can increase the inflammatory response and increase intestinal permeability and interleukin-4 levels in the intestine by activating the Th2 pathway compared to the A2 variant [86].

The difference in symptoms of milk intolerance when consuming A1 milk and A2 milk was investigated in Chinese adults [87], and it was found that gastrointestinal problems caused by consuming A2 β-casein were much lower than in the case of conventional milk. These findings indicated that A2 β-casein was associated with more severe gastrointestinal symptoms and exhibited a slight tendency for improvement of these symptoms.

Jung et al. [48] assessed the hypoallergenic property of the A2 β-casein fraction by measuring the release of histamine and TNF-α from HMC-1 human mast cells. The outcome of the study revealed no significant differences in the levels of histamine and TNF-α after treatment with A2 β-casein and in the control.

### 4.2. Effects of A2 Milk on Cardiovascular Health

Cardiovascular diseases can be caused by multiple factors and develop over a long period of time before presenting symptoms or being diagnosed. Therefore, generally, when they appear, it is very difficult to attribute them to a single cause but rather to a combination of multiple predisposing factors. The main risk factors for cardiovascular disease are high blood pressure, high LDL cholesterol and low HDL cholesterol, smoking, obesity, and physical inactivity [88]. Venn et al. [72] found that rabbits fed milk containing β-casein A2 had lower total cholesterol levels and less aortic intima-media thickening than those fed A1. Several clinical studies confirmed that the consumption of milk containing an A1 variant of β-casein favored the development of heart disease in humans. This fact was hypothesized because certain human populations that consumed milk from ruminant species free from A1 β-casein exhibited relatively lower incidences of cardiovascular diseases than Europeans [5]. At the cardiovascular level, it was reported that animals exposed to β-casein A1 showed significantly higher cholesterol, LDL, HDL, and triglyceride concentrations than subjects who consumed β-casein A2, as a control group, which consumed only whey protein [67]. These results suggest that the consumption of β-casein A1 may contribute to an increased risk of ischemic heart disease and, potentially, to a significantly higher mortality rate caused by adverse cardiovascular events.

The same findings were mimicked [67], which found that β-casein A2 was less atherogenic than A1. However, not all the work completed shows consistent results in this regard, as another randomized trial with 62 participants found no differential effects on cholesterol levels in humans consuming A1 or A2 milk or other dairy products for more than 4 weeks [72].

Similar to other BCMs, BCM-7 has been found to be related to the stimulation of LDL-cholesterol oxidation and is an important risk factor for acute cardiovascular events [11]. The LDL fraction is responsible for transporting lipids through the plasma, is essential for cholesterol transport to peripheral tissues, and is highly susceptible to oxidative damage. This oxidation plays a major role in the development of cardiovascular disease and atherosclerosis [16].

There have been several studies on the consumption of A1 and A2 β-caseins in animals, where it has been shown that those fed with A1 milk had a higher level of cholesterol and a greater deposit on the surface of the aorta. One of the most important and significant studies was performed with rabbits; Tailford et al. [67] concluded that both A1 and A2 β-caseins caused aortic fatty streaks, although A1 produced more extensive lesions. The thickness of striae in A1-fed rabbits was greater than that in A2-fed rabbits; even with increased dietary cholesterol, the thickness was significantly greater in A1 than in A2. On the other hand, the Tailford study also investigated whether β-casein A2 had an atheroprotective effect when compared with cholesterol-containing whey (0.5%). It was concluded that, for the first time, β-casein A2 had a slight protective effect, since those rabbits fed β-casein A2 at 20% had a much lower aortic thickness, a lower serum cholesterol level, and a significant reduction in LDL cholesterol [67]. Similar results were obtained in another work carried out in rabbits [64], which found that A2 is slightly less atherogenic than A1.

However, the effects of A1 or A2 milk could not be confirmed in humans. In this sense, in a randomized crossover human trial performed in New Zealand over 62 subjects, researchers found no significant differences in plasma concentrations of triacylglycerol and total LDL and HDL cholesterol between β-caseins A1 and A2 [72].

On the other hand, to reinforce this hypothesis, a posterior study was performed to examine whether A1 β-casein supplementation would promote a relative risk in people at high risk of developing cardiovascular diseases compared to A2 β-casein supplementation, but there was no evidence that this was possible [71]. People supplemented with A1 and A2 beta-casein had very similar blood cholesterol levels 12 weeks after the intervention, and aortic pressure was also similar, so it has been concluded that β-casein supplementation has no cardioprotective advantage over A1 [71]. Currently, there are not enough scientific studies that corroborate the hypothesis that β-casein A1 consumption promotes the occurrence of cardiovascular disease. 

### 4.3. Effects on Type 1 Diabetes Mellitus

Diabetes has reached epidemic levels worldwide, with almost 300 million people suffering from this disease. Although type 2 diabetes is the most common worldwide, type 1 diabetes accounts for 10–15% of all cases of diabetes and is continuously growing in all age groups [89]. Type 1 diabetes mellitus is an autoimmune disease that affects the β-cells of the pancreas, preventing them from producing insulin. Generally, it appears at early ages, which is why it is known as “childhood diabetes”; however, it can also appear in adulthood. The incidence of type 1 diabetes mellitus varies by region due to the genetic difference that exists between herds at each site, and thus the composition of their milk. Several studies have shown that the incidence of type 1 diabetes has been higher in countries such as Finland and Sweden, where A1 milk consumption is higher, while its incidence is lower in Japan and Venezuela (with the lowest A1 milk consumption per capita).

The main risk factor for type 1 diabetes is genetic, which is often triggered by environmental factors [90]. Numerous triggers have been identified, including prenatal and postnatal exposures, as is the case of β-casein A1, recognized as a key environmental trigger that may explain a significant increase in type 1 diabetes. Early exposure of infants to cow’s milk has been identified as a major factor in the development of type 1 diabetes. Autoimmunity to β-cells arises at early ages, and an early bovine milk-based diet may modulate the risk of type 1 diabetes [89].

A long-term experimental study revealed that A1 β-casein-fed mice developed diabetes after 250 days of treatment, whereas A2-fed mice were found to be nondiabetic under similar conditions [91]. Padberg et al. [92] found that in a clinical trial involving a total of 1257 individuals, increased amounts of anti-β-casein A1 antibodies were recorded among patients with type 1 diabetes and their siblings, whereas the parents and control persons contained antibodies against the A2 variant.

It has been observed that β-casein A1 affects the immune system by producing an inhibition in the lymphocytes of the intestinal system, causing an increased susceptibility of the endogenous virus that can infect the B islets of Langerhans of the pancreas and thus favoring the development of type 1 diabetes [11]. The evidence that β-casein A1 is the main trigger for type 1 diabetes is still controversial and needs to be proven. The most convincing data were provided by Laugesen and Eliott [79], who demonstrated a positive correlation between per capita consumption of β-casein A1 and type 1 diabetes in 19 developed countries. On the other hand, it has also been studied in rodent models, although the mechanisms are not well defined [89]. One has shown that generationally, the likelihood of type 1 diabetes if accompanied by A1 milk feeding increases compared to A2 milk. Four generations of rodents at risk for diabetes were fed a β-casein A1-based diet and a β-casein A2-based diet, and the results were quite conclusive, because in the first two generations, there was not much incidence of type 1 diabetes; in the next two at thirty weeks, there was a difference between those fed A1 milk and those fed A2 milk [89].

With respect to the mechanisms’ effects of β-casein regarding type 1 diabetes, it was described that BCM-7 acts as an immunosuppressant in the intestinal immune system, impairing tolerance to dietary antigens, which may contribute to the development of type 1 diabetes. It is well established that intestinal inflammation, along with accompanying changes in gut permeability, predisposes individuals to develop cellular autoimmunity by affecting gut permeability and immune activation [18]. There are two basic hypotheses for the possible association between A1 milk and type 1 diabetes. The first mechanism is based on the opioid action of the peptide product of BCM-7. The disruption of metabolic processes caused by the opioid activity of BCM-7 could result in the dysregulation of insulin and thus in an alteration of established glucose control mechanisms, including the ability of BCM-7 to reduce GSH concentrations, which can lead to cell death by ferroptosis [93], an effect that could be reversed by administering an opioid receptor inhibitor [89]. The second proposal described is through molecular mimicry of the casein protein and an epitope on the glucose transporter 2 present in the cells [89]. It is unknown whether the existence of cellular autoantibodies is only a biomarker of type 1 diabetes or a factor in the disease process [90]. In addition, it has been postulated that low expression of antioxidant enzymes may predispose islet cells to autoimmune reactions, resulting in type 1 diabetes [94]. Chia et al. [89] reviewed animal-based trials and in vitro assays and concluded that A1 β-casein and BCM-7 were the dominant triggers of type 1 diabetes in individuals with genetic risk factors. This was consistent with those reported [95], who hypothesized that exposure to A1 β-casein milk is related to the rising incidence of type 1 diabetes, and Kohil et al. [96], who postulated that BCM-7 could act as an epigenetic modulator, differentially methylating genes involved in type 1 diabetes development.

Although there are studies showing a correlation between β-casein A1 consumption and the incidence of type 1 diabetes, evidence of causality in this field is limited, and even the studies performed show contradictory results [16]. In this regard, recent work in rats found no difference between the type of milk consumed and nondifferent health markers related to diabetes [65].

A1 and A2 diets were protective in two rodent models of spontaneous type 1 diabetes, and the A1 diet was somewhat more diabetogenic [64]. However, no differences were recorded in the analysis of insulitis and pancreatic cytokine gene expression in animals fed A1 and A2. In contrast, Thakur et al. [65] found no differences in the blood profile and histopathology of the heart, liver, and kidney of diabetic rats after consuming A1 and A2 diets for 60 days.

### 4.4. Effects on Neurological Disorders

With respect to neurological disorders, a randomized double-blind clinical study with 45 participants showed that the consumption of mixed A1/A2 milk worsened cognitive processing speed and accuracy within 14 days [5], whereas subjects who consumed only A2 milk did not show this effect. Clarke and Yelland [97] found a direct relationship between the consumption of variant A1 and reduced cognitive function and, similarly, between the consumption of variant A2 and improved cognitive function in preschoolers after the consumption of A2 milk compared with conventional milk [70]. The precise mechanism or type specificity of the protein and its effects on neurological pathology are unclear, but one hypothesis is that BCM-7 interacts with the nervous system given its μ-opioid receptor stimulatory nature [11]. Another hypothesis, supported by preclinical studies in animal models, is that the A1 variant may trigger a much higher degree of inflammation than the A2 variant [16].

Opioid receptors are widely distributed throughout the brain and are also found in the spinal cord and peripheral sensory nerves and are involved in several different functions, including appetite, depression, feeding behavior, respiratory behavior, and euphoria [88]. The µ receptor to which BCMs preferentially bind shows high levels of expression in areas of the brain and spinal cord with an important role in pain and analgesia, gastrointestinal functions, or even mood and thermoregulation [88].

Among the different neurological disorders, special attention has been given to the relationship between A1 β-casein intake and autism. Autism is characterized by difficulties in the individual’s personality, social interactions, communication difficulties, and abnormal isolating behaviors [11]. It was seen to be directly related to the opioids mentioned above, as increased BCM-7 from A1 milk induces an inflammatory response in the gastrointestinal tract of children. This is exacerbated because the gut is highly permeable, leading to an increase in BCM-7 at the blood–brain barrier. This accumulation in the brain and its binding to μ-opioid receptors develop autism. This arises mostly in infants and children highly susceptible to autism due to an underdeveloped intestinal mucosa and an early introduction to their diet of cow’s milk. A direct correlation has been observed between autism and BCM-7 levels in both the blood and urine of children with autism [98]. On the other hand, it has also been observed in a recent study that A1 milk replaced by A2 milk showed an improvement in cognitive performance [11].

Sudden infant death is the most common factor seen in sudden infant death syndrome in infants within the first year of life and is the replacement of breastfeeding with bovine milk. When fed A1 milk, the milk protein is digested in the underdeveloped intestine, so BCM-7 is fully absorbed and passes directly into the bloodstream. This absorption is complete, so it passes beyond the blood–brain barrier without blocking the central nervous system. These BCM-7 molecules act as ligands for opioid receptors in the brain and can affect the respiratory system or induce depression, as well as reduce blood pressure, leading to sudden infant death [11]. Several animal studies have shown that ingestion of A1 milk causes ventilatory depression, such as sudden infant death syndrome. Ventilatory depression only occurs when BCMs are injected directly into the cerebroventricular cavity. On the other hand, in infants with such a syndrome, BCMs are found in the cerebrospinal fluid, whereas in normal infants, they are not [99]. The blood circulation of BCM-7 indicates that the complete absorption of opioid peptides from A1 milk causes neurodevelopmental alterations in humans. Another article suggested a reduction in schizophrenic symptoms related to a decrease in A1 milk intake in the USA and Europe [100].

### 4.5. Effects on the Intestinal Microbiota

The effects of the A1 and A2 β-caseins on gut microbiota have only been assayed in a mouse model thus far [63]. In this work, the A2 milk-fed group showed a higher content of fecal short-chain fatty acids (in particular, isobutyrate) of intestinal CD4+ and CD19+ lymphocytes in the intraepithelial compartment and improved villi tropism. The analysis of fecal microbiota identified Deferribacteriaceae and Desulfovibrionaceae as the most discriminant families for the A2 milk-fed mouse group, while Ruminococcaceae was the most abundant bacterial family for the A1 group. Taken together, these results suggest a positive role of milk, when containing exclusively A2 beta-casein, on gut immunology and morphology of a mouse model [63]. However, this is the only article published to date investigating the different effects of A1 and A2 β-caseins on gut microbiota, and there is very limited knowledge of the metabolomic analysis related to gut microbiota and A2 milk intake [101].

In another mouse model, it was found that A2 milk increased the levels of intestinal short-chain fatty acids produced by gut microbiota in aged mice. These fatty acids are very important for good colonic epithelial maintenance, as they are the main source of energy for colonic epithelial cells [102].

Other recent work [103] showed that A2 β-casein intake improved gastrointestinal symptoms in adults in a group of 60 volunteers, and an increase in *Bifidobacterium* spp. was found in the distal colon with respect to ordinary milk consumption. This increase in *Bifidobacterium* spp. counts was accompanied by improved symptoms of gastrointestinal discomfort, such as a reduced proportion of abdominal distension and bowel movement, increased frequency of bowel movements, and changed stool characteristics in comparison to uncharacterized ordinary milk.

There are not enough studies on the effects that the different β-caseins can cause on the human intestinal microbiota, considering that only mouse assays and only one study in humans have been performed thus far. Although the results obtained seem to suggest slight benefits at this level derived from the substitution of A1 β-caseins with A2 β-caseins, these effects must be studied from a broader point of view, since the human gut microbiota is a very complex system from which no significant conclusions can be drawn from the increase or decrease of a single genus or metabolite.

## 5. Conclusions

In general, the positive results obtained in experimental animal trials concerning the health effects of A2 milk have not always been confirmed by clinical trials in humans. However, there seems to be a sufficient consensus on the beneficial effects of A2 milk on the reduction in digestive intolerance associated with the consumption of A1 milk. This positive effect is of great importance, since a good part of the consumers who have abandoned milk consumption have done so because conventional milk causes them discomfort. Combined with the fact that some consumers are willing to pay a higher price for A2 milk, A2 milk could help to alleviate the current economic difficulties suffered by both milk producers and the dairy industry. To this end, information campaigns would be necessary, since in some countries, the level of awareness of the potential benefits of A2 milk consumption among consumers still seems to be low. For the rest of the potential positive effects on human health, despite the contradictory results to date, there have not been enough clinical trials carried out for the conclusions obtained to be definitive, and it would be very interesting to deepen the research in this sense.

In dairy producers who use a significant part of their production to produce dairy products, especially cheese, the less suitable technological properties of A2 milk compared to A1 milk must also be considered.

## Figures and Tables

**Table 2 foods-11-02387-t002:** Use of A2 milk in experimental animal trials to treat or prevent different health issues.

Animal Model	Dosage and Time of Exposure	Health Function	Main Findings	Reference
48 Wistar rats	36–48 h milk-based diets in which the β-casein component was either the A1 or A2 type	Gastrointestinal function	- A1 β-casein in rats relative to the consumption of A2 β-casein caused a delay in gastrointestinal transit. - Increased colonic myeloperoxidase dipeptidyl peptidase-4 activities.	[61]
24 male Swiss mice	Basal diet and β-casein variants (A1A1, A2A2, and A1A2) at a dose of 85 mg/animal/day suspended in 200 μL phosphate-buffered saline for 30 days	Gastrointestinal inflammation	- Consumption of A1-like variants (A1A1 and A1A2) significantly increased the levels of myeloperoxidase, inflammatory cytokines, immunoglobulin and leukocyte infiltration in intestine. - Expression for toll-like receptors (TLR-2 and TLR-4) was also upregulated on administration of A1-like variants.	[62]
24 aging Balb-c mice (20 months old)	4 weeks, with either a control diet, a diet supplemented with bovine milk containing A1/A2 β-casein (A1A2), or a diet containing A2/A2 β-casein (A2A2)	Gut morphology and histopathological alterations, gut microbiota	- Consumption of A2 milk significantly changed gut microbiota and increased content of short-chain fatty acids in gut. - Consumption of A2 milk increased content of intestinal lymphocytes in the intraepithelial compartment and improved villi tropism with respect to A1 milk.	[63]
3 different animals colonies (NOD/Ba mice, NOD/NZ mice, and BB rats)	250 days for mice and 150 for rats in which casein components were made from either β-casein A1/A1 or A2/A2 phenotype	Type 1 diabetes	- A1 or A2 β-casein diets were protected from developing diabetes. It is unlikely that diabetes could be prevented solely by removing or altering the cow’s milk component of the diet.	[64]
36 male Wistar rats	Rats were fed with respective A1 and A2 casein hydrolysate diets for 50 days. On 51st day, each group was divided into 2 subgroups (*n* = 6) and diabetes was induced in one subgroup of each group using Streptozotocin	Type 1 diabetes	- The results suggest that A1 and A2 casein hydrolysates did not have any marked effect on various health parameters.	[65]
NOD/shiLtJArc mice	Diet supplemented with A1 or A2 β-casein ad libitum during 30 weeks	Type 1 diabetes	- Dietary A1 β-casein may affect glucose homeostasis and type 1 diabetes progression.	[66]
60 rabbits	Oral diets at concentrationsof either A1- or A2 β-casein at 10%, 3.5%, or 20% concentrations for 6 weeks	Cardiovascular health	- Different cardiovascular health markers such as cholesterol, tryglicerides, LDL fatty streak lesions in the aortic arch were significantly higher in rabbits fed with A1 than for A2 β-casein.	[67]
24 male Swiss albino mice	The experimental groups were fed with basal diet and β-casein variants (A1A1, A2A2, and A1A2) at a dose of 85 mg/animal/day suspended in 200 μL phosphate-buffered saline (PBS)	Inflammatory response	- A1-like variants of β-casein induced an inflammatory response in gut by activating Th2 pathway as compared to A2 variants.	[62]
male BALB/c mice	Mice received water purified by reverse osmosis, A1A1, A1A2, and A2A2 β-casein variants of milk, respectively, at a dose rate of 10 mL/kg body weight, 5 days/week by oral gavage	Pulmonary inflammation	- Mice fed with A1 milk exhibited increased airway hyperresponsiveness with increasing concentration of bronchoconstrictor (methacholine), and inmunoglogulins, which was not observed in mice fed with A2 milk.	[68]

**Table 3 foods-11-02387-t003:** Use of A2 milk in human clinical trials to treat or prevent different health issues.

Study Design	Dosage and Time of Exposure	Health Function	Main Findings	Reference
Double-blind crossover study design in 15 patients at higher risk of developing cardiovascular disease	25 g of either β-casein A1 or A2 for 12 weeks each, with a total duration of 24 weeks	Cardiovascular health	- No evidence was found that supplementation with β-casein A1 had any cardiovascular health disadvantage over consumption of β-casein A2.	[71]
Parallel design in 67 children and adolescents aged 18 y of age, with diagnosed autism	500 mL or either A1 and A2 milk for 70 kg body weight	Neurological disorders	- Autistic children who consumed A1 milk had a 10-fold higher concentration of BCM-7 in urine than children who consumed A2 milk.	[76]
Crossover clinical trials in 26 children	Two-week period consuming at least 400 mL A1 or A2 milk each day, with a two-week washout period	Chronic functional constipation	- No significant differences were found depending on the type of milk intake.	[73]
Randomized, double-blind, crossover trial in 25 subjects	Subjects consumed a diferent randomized milk product in the morning of the day of eachvisit. Each milk meal, except the lactose-free milk, contained ~4.5 g of lactose/per 100 mL	Digestive intolerance	- Abdominal pain was lower following consumption of milk containing A2 β-casein only, compared with conventional milk.	[18]
Randomized, double-blind, 2 × 2 crossover trial in 45 subjects	14 days with a 14-day washout period at baseline and between treatment periods	Digestive intolerance	- Consumption of milk containing only A2 β-casein did not aggravate postdairy digestive discomfort symptoms relative to baseline.	[13]
Randomized, double-blinded, crossover trial in 41 women	Participants underwent a 2-week dairy washout (rice milk replaced dairy), followed by 2 weeks of milk (750 mL/day) that contained beta-casein of either A1 or A2 type, with a 2-week period of washout	Digestive intolerance	- Gastrointestinal intolerance measures such as Bristol Stool scale and abdominal pain were higher for A1 milk than A2 milk intake.	[69]
Randomized, double-blind, crossover study in 80 children	5 days consumption of 150 mL twice a day of conventional milk versus milk containing only A2 β-casein	Digestive intolerance	- Subjects who consumed milk containing only A2 β-casein had significantly less severe gastrointestinal symptoms and reduced stool frequency than those who consumed conventional milk.	[70]
Double-blind parallel design in 21 male athletes	500 mL dairy intake of A1, A2 milk, or placebo for 4 days	Muscle soreness	- No significant results were obtained regarding muscle pain recovery after exercise.	[74]
Randomized crossover trial (without washing period) in 62 subjects	Participants replaced all dairy products in their diet with 500 mL of low-fat milk and 28 g of full-fat cheese that differed in the proportion of β-casein A1 and A2 variants	Cardiovascular health	- No significant differences in serum cholesterol content were found depending on the A1 or A2 dairy products intake.	[72]
Retrospective case-control study in 55 children	Child and their mothers answered questions on breastfeeding habits and on cow’s milk products consumption	Type 1 diabetes	- Cow’s milk consumption in infancy was not found to be related to type 1 diabetes.	[75]

## Data Availability

No new data were created or analyzed in this study. Data sharing is not applicable to this article.

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
