# Peer review of "A2 Milk: New Perspectives for Food Technology and Human Health"

_foods, 2022, doi:10.3390/foods11162387_

Round 1

Reviewer 1 Report

The manuscript deals with an hot topic and could be highly attractive for researchers working in several fields (medicine, veterinary, food science, and others). In my opinion it is a good paper. However, I have the following remarks to which the authors are required to supply an answer:

- line 87: there is a repetition (the sentence is also present at line 84)

- lines 94-95: "Dairy cow breeds show different A1, A2 and B casein patterns in their milk". Is it a mistake ? Might the correct sentence be "Dairy cow breeds show different A1 and A2  pattern of B casein in their milk" ?

- line 174: its (not their)

- line 255: there's a repetition (similar sentence at lines 61-64)

- lines 268-270: "In dairy products, although BCM-7 has been detected in several types of cheeses, it did not originate from the peptides present in milk because those peptides would be removed during cheese manufacturing". The sentence is unclear: are there any BCM-T "mother peptides" in milk ? If so, please indicate them and supply references.

line 270: Nguyen et al. does not correspond to reference 37. Thus, it is not possible to check the reported information about mold cheese and semihard cheese. Please check and clarify.

Line 278: "The present review investigated the effects....". Are you meaning "Table 2 summarizes the results of the investigations about....." ?

Author Response

With respect to the comments from the Reviewer 1:

With respect to the comments about: “The manuscript deals with an hot topic and could be highly attractive for researchers working in several fields (medicine, veterinary, food science, and others). In my opinion it is a good paper. However, I have the following remarks to which the authors are required to supply an answer:”

Authors appreciate and are very grateful for the kind words from the Reviewer.

With respect to the comments about: “Line 87: there is a repetition (the sentence is also present at line 84)”

Thank you very much for your comment. The Reviewer are right and consequently, in the revised of the manuscript the repeated phrase “of which the A1 and A2 variants are the most common” was deleted from the revised version of the manuscript.

With respect to the comments about: Lines 94-95: "Dairy cow breeds show different A1, A2 and B casein patterns in their milk". Is it a mistake? Might the correct sentence be "Dairy cow breeds show different A1 and A2 pattern of B casein in their milk"?

Thank you very much for your comment. In fact it was a mistake and in order to clarify it the phrase was changed to “Dairy cow breeds show different β-casein patterns in their milk” in the revised version of the manuscript.

With respect to the comments about: “line 174: its (not their)”

Thank you very much for your comment. According to the suggestions from the Reviewer, the Word “their” was changed to “its”.

With respect to the comments about: “line 255: there's a repetition (similar sentence at lines 61-64)”

Thank you very much for your comment. According to the suggestions from the Reviewer, the phrase “For milk protein, caseins comprise 80% of milk proteins and can be divided into three major types (α-, β- and κ-caseins) [35] was deleted from the revised version of the manuscript.

With respect to the comments about: lines 268-270: "In dairy products, although BCM-7 has been detected in several types of cheeses, it did not originate from the peptides present in milk because those peptides would be removed during cheese manufacturing". The sentence is unclear: are there any BCM-T "mother peptides" in milk ? If so, please indicate them and supply references.

Thank you very much for your comment. In fact, the phrase is nuclear and was deleted from the revised version of the manuscript and changed to “is generally at lower concentrations than in milk

With respect to the comments about: line 270: Nguyen et al. does not correspond to reference 37. Thus, it is not possible to check the reported information about mold cheese and semihard cheese. Please check and clarify.

Thank you very much for your comment. In fact, the reference “Nguyen et al.” cited in this paragraph corresponds with the number 36 and not the number 37 of the list of references. It was corrected in the revised version of the manuscript.

With respect to the comments about: Line 278: "The present review investigated the effects....". Are you meaning "Table 2 summarizes the results of the investigations about..."?

Thank you very much for your comment. According to the suggestions from the Reviewer, the phrase was changed to “Table 2 summarized the results of the investigations about the effects of A1 or A2 β-casein in …”

Reviewer 2 Report

A2 milk: new perspectives for food technology and human 2 health

This review discusses the A2 milk and the new perspectives for those consumers with lactose intolerances. The review sounds interesting but there some questions and comments to consider:

Line 42: “Due  to  recent  geopolitical  events  such  as  the  Ukraine  war,  the  dairy sector crisis has been exacerbated by the increase in the costs of its operating expenses, such as fuel, energy, oils, fertilizers, and grains used for animal feed [3].”

why the authors took Ukraine war as an example of increase cost? Does this war sustain for long time? Authors might mention other reasonable reasons that really threat the dairy sector.

Line 65: “β-Casein occupies the second position in  bovine milk caseins in terms of abundance, in addition to presenting many amino  acids  [9,  10].  All  types  of  caseins  undergo  modifications  in  their structures due to the substitution or exclusion of some amino acids of the peptide chain, thus generating genetic variants.”

could you please mention how much B-CN in bovine milk as well as other casein and whey protein fractions?

Line 139-143: “This same  work  revealed  that  the  premium  price  that  Italian  consumers  are willing  to  pay  for  A2  milk  with  respect  to  fresh  lactose-free  milk  is  ap- 140 proximately  20-euro  cents/liter  [20].  Other  work  [26]  revealed  that  only  38% of a Brazilian group of consumers would pay an extra price for A2 milk with respect to conventional milk.”

are there other countries data?

Line 158: “3. Technological properties of A2 milk”

would be good to mention the physical properties of the A2 milk!

Line 384-387: “Milk allergy  occurs  mainly  in  children  under  two  years  of  age  because  their  anatomical and functional barriers and immune system are not fully developed [11].”

why would you describe something not developed yet in infants as allergy? Is this the right description?

 Line 511: “However, some authors agree that it is an added risk factor for its long-term evolution.”

Why you validate this sentence although there is not enough evidence?

Author Response

With respect to the comments from the Reviewer 2:

With respect to the comments about: “A2 milk: new perspectives for food technology and human 2 health. This review discusses the A2 milk and the new perspectives for those consumers with lactose intolerances. The review sounds interesting but there some questions and comments to consider:”

Authors appreciate and are very grateful for the kind words from the Reviewer.

With respect to the comments about: Line 42: “Due  to  recent  geopolitical  events  such  as  the  Ukraine  war,  the  dairy sector crisis has been exacerbated by the increase in the costs of its operating expenses, such as fuel, energy, oils, fertilizers, and grains used for animal feed [3].”why the authors took Ukraine war as an example of increase cost? Does this war sustain for long time? Authors might mention other reasonable reasons that really threat the dairy sector.

Thank you for your comment. In the original version of the manuscript we cited the war of Ukraine because both Ukraine and Russia are important producers of cereals used for animal feed, fertilizers and other important essential raw materials for livestock production that have become more expensive since the start of the war (especially in Europe). But it is true, we have no evidence that such a war will continue for a long time, as the reviewer says, and therefore we have removed this mention from the revised version of the manuscript.

According to the suggestion from the Reviewer, in the revised version of the manuscript we deleted the reference to the Ukraine war and included the paragraph “increment of production cost and a recent deregulation process, due to the abolition of milk quotas”

Accordingly, the referente that cited the Ukraine war as a potential cause of the Dairy sector crisis was changed to another that focuses the Dairy sector crisis on deregulation of milk production in European Union:

Pouch, T.; Trouvé, A. Deregulation and the crisis of dairy markets in Europe: Facts for economic interpretation. Stud. Political Econ. 2018, 99(2), 194-212.

With respect to the comments about: Line 65: “β-Casein occupies the second position in  bovine milk caseins in terms of abundance, in addition to presenting many amino  acids  [9,  10].  All  types  of  caseins  undergo  modifications  in  their structures due to the substitution or exclusion of some amino acids of the peptide chain, thus generating genetic variants.” Could you please mention how much B-CN in bovine milk as well as other casein and whey protein fractions?

According to the suggestions from the Revieres, the following paragraph about bovine milk proteins concentrations was added: “Among bovine milk caseins, four different types have been described: α-casein S1 (ranging 12-15 g/L), β-casein (9-11 g/L) α-casein S2 (3-4 g/L), and κ-casein (2-3 g/L). Among other protein fractions in bovine skim milk, most relevant content is for α-lactalbumin (0.6-1.7 g/L), serum albumin (0.4 m/L), immunoglobulins (0.5-0.8 g/L), lactoferrin (0.02-0.1 g/L) and secretory component (0.02-0.1 g/L) [9].”

Accordingly, it was added the following reference to the references list:

Farrell, H.M.J.; Jimenez-Flores, R.; Brown, E.M.; Buttler, J.E.; Creamer, L.K.; Hicks, C.L.; Hollar, C.M.; Ng-Kwai-Hang, K.F.; Swaisgood, H.E. Nomenclature of the proteins of cow´s milk-Sixth revision. J. Dairy Sci. 2004, 87, 1641-1674.

With respect to the comments about: Line 139-143: “This same  work  revealed  that  the  premium  price  that  Italian  consumers  are willing  to  pay  for  A2  milk  with  respect  to  fresh  lactose-free  milk  is  approximately  20-euro  cents/liter  [20].  Other  work  [26]  revealed  that  only  38% of a Brazilian group of consumers would pay an extra price for A2 milk with respect to conventional milk.” Are there other countries data?

It is difficult to asses a provide accurate information on the price difference between A2 milk and conventional milk, as A2 milk is often sold with other claims that increase its price, such as "welfare", "organic", etc. To the best of our knowledge, the cited premium prices stated in the text were the only that we found in the scientific literature. However, to complete the information requested by the Reviewer, we included in the revised version of the manuscript an information found in the website of the first company to commercialize this type of milk (The A2 milk company). It was included in the text the following paragraph “The first dairy company commercializing A2 milk (The A2 milk company) claims that they pays a premium of around 5-7 % to its farmer suppliers in New Zealand, Australia, and the United Kingdom [28].”

With respect to the comments about: Line 158: “3. Technological properties of A2 milk” would be good to mention the physical properties of the A2 milk!

According to the suggestion from the Reviewer, the included data about physical properties of A2 milk, as well as aminoacid and fat content provided by a very recent article. In the revised version of the manuscript is was included the paragraphs: “A recent work [28] investigating the sensory, colour and composition of A2 milk comparing with A1 milk found that different genotypes did not affect the smell, taste or general acceptance of the milk. However, some differences were found in colour. A2 milk showed colour paramethers more close to colour gold standard for colour, making it more appealing to consumers without artificial food colouring [28].”; And “A recent work found slight differences in aminoacid composition, of A2 and A1 milk, showing A2 milk higher amount of leucine that A1 milk but lower overall amino acid content [28].”

Additionally the following new referente was included in the revised version of the manuscript:

de Vitte, K.; Kerziene, S.; Klementaviciute, J.; de Vitte, M.; Miseikine, R.; Kudlinskiene, I.; Cepaité, J.; Dilbiene, V.; Stankevicius, R. Relationship of β-casein genotypes (A1A1, A1A2 and A2A2) to the physicochemical composition and sensory characteristics of cow´s milk. J. Appl. Anim. Res. 2022, 50(1), 161-166.

With respect to the comments about: Line 384-387: “Milk allergy  occurs  mainly  in  children  under  two  years  of  age  because  their  anatomical and functional barriers and immune system are not fully developed [11].” Why would you describe something not developed yet in infants as allergy? Is this the right description?

In fact, this is not a good definition of allergy. In the revised version of the manuscript, we deleted it.

With respect to the comments about: Line 511: “However, some authors agree that it is an added risk factor for its long-term evolution.” Why you validate this sentence although there is not enough evidence?

We were not validating, but only reflecting the opinion of previous published work. However, we consider that the Reviewer is right and consequently, we deleted the phrase “However, some authors agree that it is an added risk factor for its long-term evolution.” from the revised version of the manuscript.

Round 2

Reviewer 2 Report

Thanks for the authors for considering the comments raised in the first revision. Please adjust the page #s in through the paper.